# TRACEABLE FEDERATED CONTINUAL LEARNING

## ABSTRACT

Federated continual learning (FCL) is a typical mechanism to achieve collaborative model training among clients that own dynamic data. While traditional FCL methods have been proved effective, they do not consider the task repeatability and fail to achieve good performance under this practical scenario. In this paper, we propose a new paradigm, namely *Traceable Federated Continual Learning (TFCL)*, aiming to cope with repetitive tasks by tracing and augmenting them. Following the new paradigm, we develop **TagFed**, a framework that enables accurate and effective **T**racing, **aug**mentation, and **Fed**eration for TFCL. The key idea is to decompose the whole model into a series of marked sub-models for optimizing each client task, before conducting group-wise knowledge aggregation, such that the repetitive tasks can be located precisely and federated selectively for improved performance. Extensive experiments on our constructed benchmark demonstrate the effectiveness and efficiency of the proposed framework. Source code will be released after notification.

## 1 INTRODUCTION

With the popularity of federated learning (FL) (McMahan et al., 2017), more and more researchers are devoted to applying FL to a variety of applications such as object detection (Liu et al., 2020), medical image analysis (Liu et al., 2021c), and recommendation systems (Yang et al., 2020). Despite being effective, most of the existing FL methods assume that client data are stable and will not be changed over time. However, in real-world applications, data may come continuously and we cannot predict the data variability in advance. The research community has noticed this problem and proposes a new concept called federated continual learning (FCL) (Yoon et al., 2021; Dong et al., 2022), where each local client receives data in a streaming manner and a central server attempts to achieve federation in such a dynamic environment.

We notice that current FCL solutions pay more attention to the typical problems existed in CL such as catastrophic forgetting (Dong et al., 2022). However, all of them obey the following unpractical data setting: *the upcoming task data are completely different from previous tasks*. We argue that in real-world scenarios, similar or even identical task data have a large possibility to appear many times. **On the one hand**, the client itself is full of uncertainty. When the FCL process runs for a long time, it is inevitable to encounter repetitive tasks, especially for the federated scenario that may include a large number of clients. **On the other hand**, previous tasks may have limited data in that time and cannot achieve good performance. Therefore, the developers have the motivation to further improve the performance by collecting more data for that task in the future. In a word, the repetitive task sequence is commonly seen and should be regarded as a default setting.

Unfortunately, to the best of our knowledge, existing FCL techniques do not take the task repeatability into consideration and fail to achieve good performance under the condition (see in Section 3.1). To fill the gap, in this paper, we propose a new paradigm, namely **Traceable Federated Continual Learning (TFCL)**, which aims at *tracing the previous repetitive data features to the current one and augmenting them before federation, in order to further boost the performance.* As shown in Figure 1, for each client, instead of treating each incoming task as equal like the traditional FCL does, TFCL checks whether the current task is a repetitive task, which will be traced and connected to the previous target task for further processing before federation. Intuitively, compared to the traditional continual learning in the client side, the proposed paradigm decreases the number of tasks since the repetitive task is not considered as a new task, which reduces the loss caused by new task learning and benefits the overall performance.

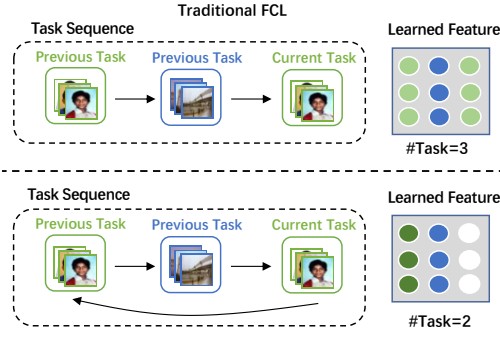

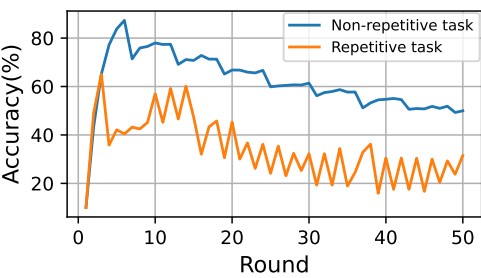

Figure 1: Comparison of traditional FCL and traceable FCL in one client. Here different colors denote different features. The white color means that the neuron has no feature (i.e., unused). The darker color indicates the stronger feature representation ability.

Figure 2: Performance of the traditional FCL method GLFC on the non-repetitive and repetitive task settings. Here the accuracy refers to the whole task performance as the learning continues.

Achieving TFCL is non-trivial and requires overcoming the following key challenges. First, the client cannot store all the previous task data due to the limited storage, which means that the trained model in the current state is the only resource we can utilize to infer and augment the previous features. Considering the poor interpretability of neural networks, it is challenging to accurately trace and optimize the corresponding data features. Second, the task repeatability in different clients may be irregular, which suggests that we cannot directly federate them because in a certain timeline, the feature distribution of repetitive tasks among clients might be highly heterogeneous. Therefore, a novel federation scheme is needed to cope with the heterogeneous situation.

In this paper, we address the aforementioned challenges by proposing **TagFed**, a framework that enables accurate and effective **T**racing, **aug**mentation, and **Fed**eration for TFCL. The rationale behind TagFed is to (1) decompose the whole model into a series of marked sub-models to selectively optimize different client task; (2) conduct group-wise knowledge aggregation for efficient federation. Specifically, to tackle the first challenge, we present *traceable task learning*, where the model pruning technique (Han et al., 2015) is utilized to iteratively generate sub-models for different client tasks. Note that each new task can only use the neurons that are not occupied for previous tasks to learn the corresponding features. Besides, we mark the weight of each task and save a weight copy of previous repetitive tasks. This allows selective retraining of specific weights for the current repetitive task, while still utilizing weight copies for non-repetitive tasks. In this way, when a repetitive task comes, we can accurately locate its corresponding features by the marked neurons, which will be easily retrained as augmentation, without introducing extra learning parameters like a new task.

TagFed addresses the second challenge by designing *group-wise knowledge aggregation*, whose key idea is to construct several model groups in the server and federate knowledge in a client-server transfer manner. For similar or identical repetitive tasks, their client knowledge is sequentially transferred to a same server model (i.e., a group) as aggregation and the server model then puts the aggregated knowledge back to each client. Here we do not use the head-to-head aggregation methods (e.g., FedAvg (McMahan et al., 2017)) because the model parameters of previous tasks are frozen and cannot be modified.

Considering that there is no available benchmark to conduct TFCL, we manually construct a benchmark based on public datasets (details in the appendix), where we simulate the repeatability range and degree for different clients. We evaluate our framework on the benchmark and compare it to the traditional FCL baselines. The results show that our approach significantly outperforms other methods on accuracy performance by a large margin while using roughly 50% communication cost. In addition, we conduct a series of in-depth theoretical analyses (details in the appendix) and empirical experiments to prove the effectiveness of the proposed framework.

This paper makes the following contributions:

- We propose a new FCL paradigm, called Traceable Federated Continual Learning (TFCL), where previous tasks can be traced and further optimized to deal with the repetitive task sequence in an FL system. To the best of our knowledge, this is the first attempt in the literature to study and explore TFCL.

- We design and implement TagFed, a framework to enable accurate and effective TFCL. Through feature tracing, sub-model augmentation, and group-wise knowledge federation, TagFed can achieve higher model accuracy at a small communication cost.

- Extensive experiments on our constructed benchmark demonstrate the superiority of TagFed over other baselines.

## 2 RELATED WORK

### 2.1 CONTINUAL LEARNING

Continual learning (CL) aims to learn deep models from the data that gradually come in a sequence during the training phase. Recent continual learning approaches mainly include two categories: task-based CL, where all class data come but are from a different dataset for each phase (Chaudhry et al., 2018; 2019; Davidson & Mozer, 2020); class-based CL, which means that each phase receives the data of a new set of classes from an identical dataset (Castro et al., 2018; Hou et al., 2019; Hu et al., 2021). The latter one is typically called class-incremental learning (CIL) and our work follows this data setting. Specifically, most of the related methods pay more attention to solving the problem of forgetting old data. For instance, Mallya *et al.* (Mallya & Lazebnik, 2018) packed multiple tasks into a single model by pruning the unimportant neurons. Douillard *et al.* (Douillard et al., 2020) proposed a spatial-based distillation loss applied throughout the model and a representation comprising multiple proxy vectors for each object class to enhance the performance. In our work, we are devoted to extending traditional CL under the federation scenario and propose a new paradigm, namely *traceable federated continual learning*, to address the task repeatability problem in a streaming data sequence.

### 2.2 FEDERATED LEARNING

Federated learning (FL) has been proposed as a privacy-preserving distributed machine learning approach that enables training on a large corpus of decentralized user data (McMahan et al., 2017; Hard et al., 2018; Yang et al., 2018; 2019; Oh et al., 2022). Much effort has been made in this hot research topic, including attacks/defense on FL (Melis et al., 2019; Bhagoji et al., 2019; Li et al., 2019; Yang et al., 2022), personalized FL (Yu et al., 2020; Yang et al., 2021; Liu et al., 2021b), data heterogeneity on FL (Sim et al., 2019; Mansour et al., 2020; Liu et al., 2021a), communication-efficient FL (Konečnỳ et al., 2016; Ji et al., 2020), etc. Recently a few works have begun to explore FL on dynamic client data, which are called *federated continual learning (FCL)*. For example, Yoon *et al.* (Yoon et al., 2021) proposed FedWeIT, a framework that can achieve effective FCL by alleviating the interference between incompatible tasks and boosting the positive knowledge transfer across clients during learning. Dong *et al.* (Dong et al., 2022) proposed a practical scenario, called Federated Class-Incremental Learning (FCIL), where each local client collects data continuously with its own preference and allows new clients with unseen new classes to join in the FL training at any time. Luopan *et al.* (Luopan et al., 2023) proposed FedKNOW, aiming at an accurate and scalable federated continual learning via a novel signature task knowledge. Different from the above FCL approaches assuming that the task sequence among clients is non-repetitive, our framework manages to effectively cope with the task repeatability and achieve traceable FCL.

## 3 PROBLEM FORMULATION

### 3.1 FEDERATED CONTINUAL LEARNING

Generally, the standard FCL process is composed of the following key steps: (1) The client side copes with the streaming tasks with a specially designed loss for typical continual learning (Shmelkov et al., 2017; Li & Hoiem, 2017); (2) At a timestep, the continually learned models in each client

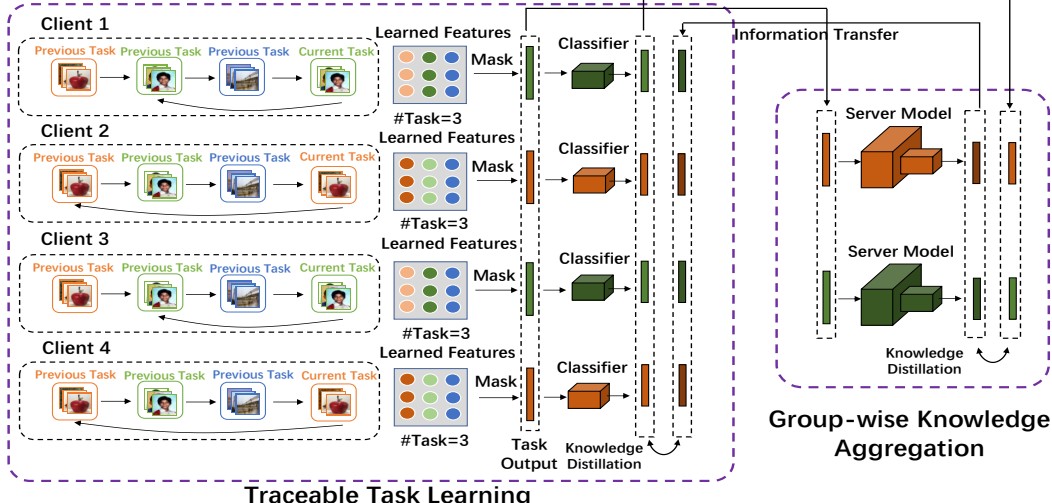

Figure 3: Overview of the proposed TagFed framework. It mainly includes two modules: *traceable task learning* and *group-wise knowledge aggregation*. Here different colors represent different features and the darker the color is, the more sufficient features the neuron holds.

are uploaded to a central server, where different model aggregation methods are implemented to generate an improved global model. We iterate this pipeline many rounds until model convergence. Compared to traditional federated learning, FCL assumes that the overall system runs in a dynamic data environment, which is more challenging to achieve effective federation. Recent solutions (Yoon et al., 2021; Dong et al., 2022) attempt to address the client interference and catastrophic forgetting problem that are widely appeared in FCL, without considering the data repeatability in a task sequence.

To observe the influence of the repetitive task environment, we conduct a preliminary experiment, where we employ a state-of-the-art FCL method GLFC (Dong et al., 2022) to evaluate the performance on the non-repetitive and repetitive task settings respectively. Here CIFAR100 (Krizhevsky et al., 2009) is used and partitioned to simulate 10 clients and we run GLFC for 50 federated rounds. As shown in Figure 2, we can clearly see that after introducing the repetitive tasks, the performance degradation is obvious, which means that the typical FCL methods fail to achieve effective local task learning and global knowledge aggregation.

## 3.2 TRACEABLE FEDERATED CONTINUAL LEARNING

Motivated by the poor performance of current FCL solutions in the repetitive task environment, we attempt to introduce a new paradigm, *Traceable Federated Continual Learning*, to achieve effective FCL under the repetitive data stream. Specifically, our data setting differs previous ones from two aspects: (1) For a certain time, the incoming task has ever appeared in the previous task sequence; (2) Besides the task repeatability, the incoming task in different clients can also be various.

Formally, assuming there are $N$ local clients, each of which contains a task sequence $S_t^i = \{s_1^i, s_2^i, ..., s_n^i\}$ at the time $t$. Here $i$ denotes the $i_{th}$ client and $n$ is the number of the task sequence. If $S_{t-k}^i = S_t^i$, we believe a repetitive task appears at the time $t$ and should be traced back to the time $t - k$ for reprocessing. Based on the symbols, we define the goal of *Traceable Federated Continual Learning* as follows.

**Definition 3.1. (Traceable Federated Continual Learning (TFCL))**. *Suppose the current model $M_{cur}$ has learned $P = \{p_1, p_2, ..., p_q\}$ tasks, which are reflected by corresponding features $F = \{f_1, f_2, ..., f_q\}$. Here $q$ is the number of previous tasks. When the incoming task $p_j \in P$, the goal of TFCL is to accurately trace and connect the corresponding data features $f_j$ to $p_j$ for further processing, and then selectively federate the various clients for an improved model $M_{new}$.*

In the following sections, we will describe a series of designed techniques in detail for accomplishing our objective.

# 4 FRAMEWORK DESIGN

## 4.1 OVERVIEW

We design and develop TagFed, a framework to achieve TFCL via tracing, strengthening and selectively federating previous certain features, such that the learning process will not be interfered by the repetitive task. Figure 3 depicts the key modules of TagFed, which can be briefly summarized as follows.

- *Traceable Task Learning (TTL)*: This module takes advantage of the pruning technique to obtain incremental sub-models for incoming tasks. When a repetitive task comes, instead of processing it as a new task, the designed module traces the corresponding features of previously marked sub-models and further retrains the parameters for learning more sufficient features (the darker neurons in the figure). The learned model in each client is then masked to leave the current data features for later federation.

- *Group-wise Knowledge Aggregation (GKA)*: This module constructs a client-server knowledge transfer pipeline as federation. The key message between the two ends is a series of data features and logits, which comes from different clients. Considering the variety of the uploaded data features, we develop multiple groups in which tasks with similar or identical features are fed into a same server model as knowledge aggregation. The output of the aggregated server model is finally transferred back to corresponding clients to provide more valuable knowledge.

The two modules will be operated many times until we achieve desirable performance. In the remainder of the section, we describe in detail our approach for implementing each module.

## 4.2 TRACEABLE TASK LEARNING

The basic idea for our *Traceable Task Learning (TTL)* is employing a sub-model to train each task data and freezing the used weights during new task training. When a new task comes, we compare the distribution with previous ones that have been reserved to figure out whether it is a repetitive task. If there is no clear boundary, we consider it as a new task to avoid any possible negative impact on previous tasks.

**Principle of sub-model generation.** Assume that all clients begin with a shared model $M_{ini}$ and the $i_{th}$ client is learning for task $p_i$. The initial model is first trained via its local dataset and then each client performs gradual pruning (Zhu & Gupta, 2017) on the model, which removes a portion of the weights to obtain a sub-model and fine-tunes the sub-model iteratively to maintain accuracy. Here the size of each sub-model is based on the task complexity and can be set by each client. After obtaining the sub-model, we keep its weights unalterable to avoid forgetting and enable learning until a repetitive task $p_i$ comes to the $i_{th}$ client in the future. According to the principle of the sub-model generation, in the following parts, we respectively illustrate how to cope with non-repetitive and repetitive new tasks.

**Incremental training for non-repetitive new tasks.** Suppose current model $M_{cur}$ has learned $P = \{p_1, p_2, ..., p_{q-1}\}$ tasks, task $p_q$ is coming and $p_q \neq p_i, i = 1, 2, ..., q - 1$. The model weights preserved for task $p_1$ to task $p_{q-1}$ are denoted as $W_{1:q-1}^P$ (i.e., a series of sub-models). The pruned weights associated with task $p_{q-1}$ are denoted as $W_{q-1}^E$ that can be used to learn new tasks. We apply a learnable mask $B_q \in \{0, 1\}^D$ to pick the old weights $W_{1:q-1}^P$, where $D$ is the dimension of $W_{q-1}^E$. The picked weights are represented as $B_q \odot W_{1:q-1}^P$, where $\odot$ is the element-wise product between $B$ and $W_{1:q-1}^P$. We then learn a real-valued mask $\hat{B}_q$ and applies a threshold for binarization to generate $B_q$. Specifically, when training the binary mask $B_q$, we update the real-valued mask $\hat{B}_q$ in the backward pass and quantize it with a threshold on $\hat{B}_q$. Hence, the model can pick a part of $W_{1:q-1}^P$ to re-use for the new task via the mask. Besides, the weights $W_{q-1}^E$ can be used for task $p_q$.

The mask $B$ and $W_{q-1}^E$ are optimized together on the training data of task $p_q$ with the loss function via back-propagation. Then we will gradually prune the model once again. When the ratio of weights for task $p_q$ meets the task requirement, they are represented as $W_q^P$ and frozen to get rid of the old task's forgetting. This incremental training process will be iterated many times until there are no non-repetitive new tasks.

**Traceable augmentation for repetitive new tasks.** Suppose current model $M_{cur}$ has learned $P = \{p_1, p_2, ..., p_{q-1}\}$ tasks, task $p_q$ is coming and $p_q = p_i = ... = p_j, q > i > ... > j$. The model weights preserved for task $p_i$ and $p_j$ are denoted as $W_i^P$ and $W_j^P$. The real value mask of task $p_i$ is represented as $\hat{B}_i$. We achieve traceable augmentation by (1) saving a copy of $W_j^P$ to prevent the influence from other tasks; (2) retraining the real value mask $\hat{B}_i$ and $W_i^E$ together on the training data of task $p_q$ with the loss function via back-propagation and finally generating $\hat{B}_q$ and $W_q^P$. Note that $B_q$ depending on $\hat{B}_q$ picks a part of $W_{1:q-1}^P$ and until the next repetitive task $p_q$ comes, we will apply $W_q^P$ to process task $p_q$. Otherwise, we will apply $W_j^P$ to keep the accuracy of other tasks.

**Difference to other CL methods.** Although the pruning-based CL methods can also be used to trace the repetitive tasks, it can only pinpoint the relevant weights and fine-tune them, which may affect the performance of non-repetitive tasks, as these tasks also rely on the frozen weights. In our approach, we devise a copying mechanism to store the weights frozen previously, along with marking their repetition frequency. This enables selective utilization for both repetitive and non-repetitive tasks.

## 4.3 GROUP-WISE KNOWLEDGE AGGREGATION

*Group-wise Knowledge Aggregation (GKA)* serves as the second step to achieve effective federation among clients. We cannot use the traditional aggregation approach in FL (e.g., FedAvg (McMahan et al., 2017)) because they attempt to change the weights of the whole model, which is infeasible for our client model that is divided into several sub-models, whose weights are frozen to avoid forgetting previous tasks when learning new tasks. Therefore, we construct a client-server bridge for information transfer and introduce group-wise knowledge distillation to accomplish aggregation.

**Information transfer.** In our framework, clients require extracting the feature maps after a specific hidden layer and related logits in their local models, which will be sent to the server during training. Note that the feature maps may leak users' confidential information, we can add some designed noise to further ensure privacy (details in the appendix). The server side will then use different models to aggregate knowledge from uploaded information according to current features of the client task and send the corresponding predicted value back to clients as the aggregated knowledge. Based on the knowledge, clients will further retrain and distill their models for improved performance.

**Server-side knowledge distillation.** In our TFCL, the feature distribution of repetitive tasks among clients can exhibit significant heterogeneity within a given timeframe. This circumstance has driven us to create distinct groups to facilitate selective knowledge distillation rather than blindly distilling them together. Given a series of feature maps and logits, we first put the feature maps into different groups based on the logits and construct a server model for each group. The number of server models depends on the number of learning tasks. In this way, the knowledge of identical tasks can be clustered for effective aggregation, getting rid of the interference from other irrelevant tasks. Concretely, for each group, the server loss function is defined as:

$$\ell_s = \alpha_s \ell_{CE} + \beta_s \ell_{KD} \left( \boldsymbol{z}_s, \boldsymbol{z}_c^{(k)} \right)$$
$$= \alpha_s \ell_{CE} + \beta_s D_{KL} \left( \boldsymbol{p}_k \| \boldsymbol{p}_s \right)$$

$$(1)$$

where $\ell_{CE}$ is the cross-entropy loss between the predicted values and the ground truth labels. $D_{KL}$ is the Kullback Leibler (KL) Divergence function. $\boldsymbol{z}_s$ and $\boldsymbol{z}_c^{(k)}$ are the predicted values of server models and client models. $\boldsymbol{p}_k^i = \frac{\exp\left(z_c^{(k,i)}/T\right)}{\sum_{i=1}^C \exp\left(z_c^{(k,i)}/T\right)}$ and $\boldsymbol{p}_s^i = \frac{\exp\left(z_s^i/T\right)}{\sum_{i=1}^C \exp\left(z_s^i/T\right)}$, where $C$ is the number of clients and $T$ is a temperature parameter. They are the probabilistic prediction of the $k_{th}$ client model and the server model. $\alpha_s$ and $\beta_s$ are the hyperparameters to control the ratio of the cross-entropy loss and the Kullback Leibler Divergence. Note that in each federation round, the server will receive the message sent from clients and sequentially distill the corresponding server model based on Eq. 1 for each group.

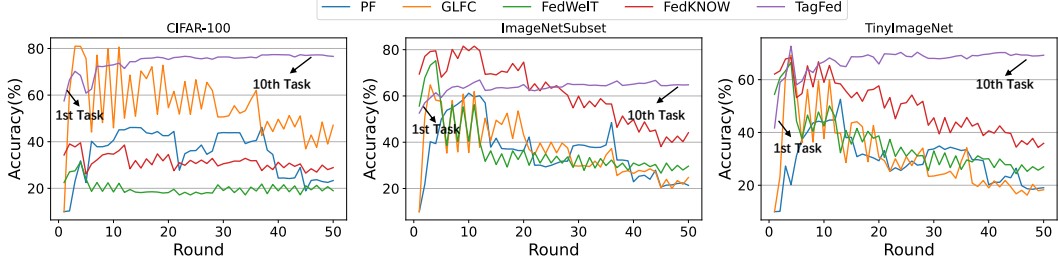

Figure 4: Convergence comparison on different datasets. Here the final accuracy of different methods is represented at the location of "10th Task" since we run 10 tasks in this FCL process. *PF* denotes *PackNet+FedAvg*.

**Client-side knowledge distillation.** Different from the distillation on the server side, the client-side knowledge distillation does not need to maintain several models for training. Instead, they connect to the corresponding server model based on the logits and utilize its aggregated knowledge to improve the local performance. For each client, the optimization loss can be defined as:

$$\ell_c^{(k)} = \alpha_c \ell_{CE} + \beta_c \ell_{KD} \left( \boldsymbol{z}_s, \boldsymbol{z}_c^{(k)} \right)$$
$$= \alpha_c \ell_{CE} + \beta_c D_{KL} \left( \boldsymbol{p}_s \| \boldsymbol{p}_k \right) \quad (2)$$

where most of the symbols have been formally defined in Eq. 1 except $\alpha_c$ and $\beta_c$, which are another two hyperparameters to control the loss in the client end. As a result, each client can benefit from the valuable knowledge aggregated by the corresponding server model.

## 5 EXPERIMENTS

### 5.1 EXPERIMENTAL SETUP

**Benchmark.** Since there is no available benchmark to evaluate the performance of our framework, we manually construct one with three datasets that are widely used in the field of federated continual learning: CIFAR-100 (Krizhevsky et al., 2009), ImageNetSubset (Deng et al., 2021) and TinyImageNet (Le & Yang, 2015). As stated in Section 2.1, our settings mainly follow class-incremental learning (CIL), where each class is regarded as a task. The concrete description of the repetitive task settings and other implementation details can be found in the appendix.

**Baselines.** To the best of our knowledge, there are few works targeting federated continual learning and most of the baselines just combine continual learning and federated learning directly. Here we select one representative combination: PackNet (Mallya & Lazebnik, 2018) + FedAvg (McMahan et al., 2017). Besides, we choose FedWeIT (Yoon et al., 2021), GLFC (Dong et al., 2022) and FedKNOW (Luopan et al., 2023) as baselines, which are state-of-the-art methods specially designed for federated continual learning. In the appendix, we provide a detailed description of these baselines and other hyperparameter settings.

### 5.2 PERFORMANCE COMPARISON

**Accuracy comparison on our benchmark.** As shown in Figure 4, we can clearly see that the proposed framework outperforms other methods by a large margin in the final accuracy. This demonstrates that by traceable task learning and group-wise knowledge aggregation, we can greatly boost the accuracy performance for the task repeatability scenario. Besides, it is worth noting that: (1) *GLFC* performs well at the beginning while degrading dramatically as new tasks come, which indicates that the repetitive tasks disturb its original learning state. This phenomenon also appears in *FedWeIT* and *FedKNOW* except the CIFAR-100 dataset. The reason may be that the two methods can easily become overfitting when the data scale is small such as CIFAR-100; (2) It is obvious that *PackNet+FedAvg* obtains poor performance in most of the learning process. We believe this is due to the incompatibility between the weight-level aggregation and the dynamic data settings; (3) In addition to the final accuracy, the convergence speed of TagFed is faster than others, which means that our framework can quickly adapt to a new repetitive or non-repetitive task.

Table 1: Effect of the proposed modules of different task numbers on CIFAR-100. *Ours-w/oModule* denotes the performance of our framework without using the Module. Here "Task Scale" refers to the number of tasks.

| Task Scale | 2 | 4 | 6 | 8 | 10 | Avg |
|---|---|---|---|---|---|---|
| *Ours-w/oTTL* | 69.20% | 59.05% | 55.62% | 55.65% | 56.05% | 59.11% |
| *Ours-w/oGKA* | 71.00% | 73.43% | 75.25% | 76.21% | 75.76% | 74.33% |
| *Ours* | **71.40%** | **75.39%** | **76.36%** | **77.18%** | **78.35%** | **75.73%** |

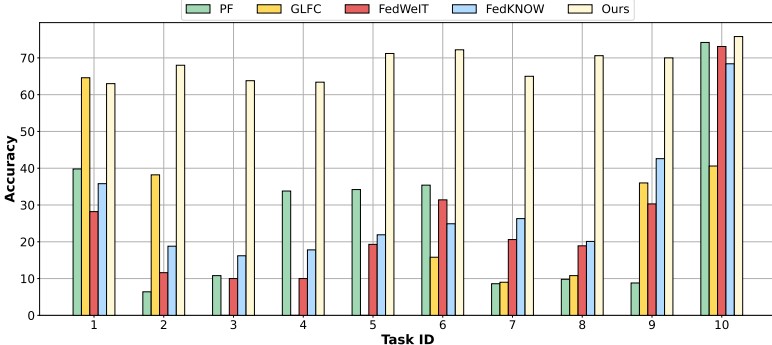

Figure 5: The task-specific performance on the TinyImageNet dataset.

**Ablation study.** TagFed mainly includes two key components: traceable task learning (TTL) and group-wise knowledge aggregation (GKA). Here we conduct ablation studies to explore whether each of them is beneficial to the final performance. Table 1 summarizes the results, where we record the corresponding accuracy performance of different task scales on CIFAR-100. From the table, we can find that both the two modules are essential to cope with the TFCL scenario.

**Task-specific performance.** In addition to the overall performance on all tasks, we are also interested in the detailed performance on each task. Figure 5 records the task-specific performance on the TinyImageNet dataset, where we set 10 tasks and each of which includes 20 classes. From the figure, we can observe that the accuracy on different tasks is various for all the methods. Specifically, *PackNet+FedAvg* has low performance on the first several tasks while achieving excellent performance on the final task. We believe this is because *PackNet+FedAvg* pays more attention to

Table 2: Results on the noise-introduced situation with CIFAR-100. Here *PF* denotes *PackNet+FedAvg*.

| Task scale | 2 | 3 | 5 |
|---|---|---|---|
| *PF* | 45.30% | 42.96% | 31.66% |
| *GLFC* | 50.10% | 54.57% | 62.02% |
| *FedWeIT* | 18.24% | 18.17% | 18.86% |
| *FedKNOW* | 36.17% | 34.50% | 31.87% |
| *Ours(1/dFIL = 5)* | 70.30% | 73.06% | 73.38% |
| *Ours(1/dFIL = 1)* | 70.50% | 73.86% | 73.60% |
| *Ours(1/dFIL = 0.5)* | 71.20% | 72.93% | 74.10% |
| *Ours(w/o noise)* | 71.40% | 75.27% | 75.99% |

the current task and federates them without considering the task forgetting problem, which will obtain poor accuracy in the previous tasks. Other FCL baselines, however, despite addressing catastrophic forgetting, fail to deal with the repetitive tasks and might introduce a large new task learning loss as the repetitive task increases. As a result, they cannot reach good performance and even for some tasks, the accuracy becomes zero. The results on other datasets can be found in the appendix.

**Noise-introduced performance.** Because our pipeline requires exchanging the feature maps between the clients and the server, we introduced a designed noise to further ensure privacy. Here we adopted *diagonal Fisher information leakage* (dFIL) as the privacy metric. The detailed noise setting can be found in the appendix. As shown in Table 2, although the performance of ours decreases slightly compared to the version without noise, it still significantly surpasses other baselines, which validates the effectiveness of TagFed under the privacy-preserving scenario.

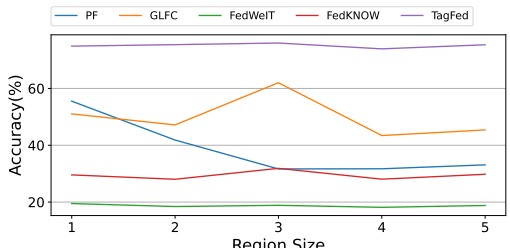 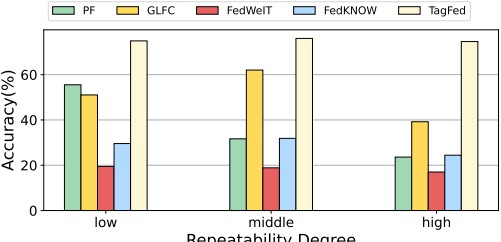

Figure 6: (a) Performance on the non-iid condition. (b) Effect of the Repeatability Degree.

Table 3: The number of parameters needed for communication and computation on different methods.

| Method | Uploading Cost | Offloading Cost | Computational Cost |
|---|---|---|---|
| *PackNet+FedAvg* | 11.70MB | 11.70MB | 6.33 MB * 40 rounds |
| *GLFC* | 11.80MB | 11.70MB | 11.50 MB * 45 rounds |
| *FedWeIT* | 9.70MB | 26.20MB | 14.70 MB * 21 rounds |
| *FedKNOW* | 11.70MB | 11.70MB | 11.50 MB * 32 rounds |
| *Ours* | 10.30MB | 0.05MB | 11.50 MB * 15 rounds |

**Performance on the non-iid condition.** The non-iid degree can be represented by the backtracking region, which decides the number of previous tasks to trace. The larger region we backtrack, the higher heterogeneity we are in. Here we set the region size from 1 to 5, with CIFAR-100 running for 5 tasks to observe the performance. As shown in Figure 6 (a), TagFed can achieve consistently better performance regardless of the heterogeneity degree.

### 5.3 EFFICIENCY OF TAGFED

In this subsection, we study the efficiency of TagFed, where we record the uploading, offloading, and training parameters as the cost measurement. Table 3 demonstrates the results. For the uploading cost, we can see that the difference among these methods is not obvious, which means that the cost of uploaded feature maps is comparable to the whole model. For the offloading cost, TagFed can show a great advantage over other methods since it only needs the computed prediction vector, whose dimension is extremely small. Besides, we can see that TagFed does not result in increased computation cost as most training parameters have been frozen in the later FL rounds. Moreover, due to its fast convergence speed, we can complete FL with fewer rounds, further reducing the overall computation cost. In general, our framework can save roughly 50% communication cost compared to others while saving a little computational cost.

### 5.4 EFFECT OF THE REPEATABILITY DEGREE

This subsection explores the effect of the repeatability degree on the final performance. Here we partition the repeatability degree into three levels: *low repeatability*, *middle repeatability*, and *high repeatability*, in terms of tracing rounds and regions (details in the appendix). We evaluate different methods on CIFAR-100 with five tasks. As shown in Figure 6 (b), unlike other baselines that are significantly influenced by the degree of repeatability, TagFed is not affected and achieves improved performance. This further validates the effectiveness of TagFed in handling repetitive tasks.

## 6 CONCLUSION

In this paper, we propose Traceable Federated Continual Learning (TFCL), where repetitive tasks can be accurately and effectively traced and processed to boost the performance. We design and implement TagFed, a framework to achieve TFCL through feature tracing, sub-model augmentation, and group-wise knowledge federation. Experiments on our simulated benchmark demonstrate the effectiveness and efficiency of TagFed, which outperforms other state-of-the-art methods while saving communication and computational cost.

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

Table 4: Statistics of our simulated benchmark.

| Dataset | # Simulated Clients | # Training Samples | # Testing Samples |
|---|---|---|---|
| *CIFAR-100* | 10 | 50000 | 10000 |
| *ImageNetSubset* | 10 | 50000 | 10000 |
| *TinyImageNet* | 10 | 45000 | 5000 |

# A  DETAILED EXPERIMENTAL SETTINGS

## A.1  CONSTRUCTED BENCHMARK

We manually construct a benchmark with public datasets used for FCL (Dong et al., 2022) to simulate the task repeatability scenario. The detailed statistics are illustrated in Table 4 and we briefly introduce the setting of each dataset as follows.

- *CIFAR-100 (Krizhevsky et al., 2009)*: CIFAR-100 is a classification dataset, which owns 100 classes and each class contains 600 images. There are 500 training images and 100 testing images per class. The 100 classes in CIFAR-100 are grouped into 20 superclasses. Each image comes with a "fine-grained" label (the class to which it belongs) and a "coarse-grained" label (the superclass to which it belongs).

- *ImageNetSubset (Deng et al., 2021)*: The mini-ImageNet dataset is widely used for few-shot learning evaluation. Its complexity is high due to the use of ImageNet images while requiring fewer resources and infrastructures than directly running on the full ImageNet dataset. In total, there are 100 classes with 600 samples of 84×84 color images per class.

- *TinyImageNet (Le & Yang, 2015)*: TinyImageNet is a subset of the ILSVRC2012 classification dataset. It consists of 200 object classes, each of which provides 500 training images, 50 validation images, and 50 test images. All images have been downsampled to $64 \times 64 \times 3$ pixels.

To make a fair comparison, we follow the typical client settings in FCL (Dong et al., 2022). For the task repeatability settings, we divide federating rounds into ordinary rounds and backtracking rounds. Client models will learn the current task in an ordinary round and learn repetitive tasks in a backtracking round. For each backtracking round, the incoming tasks of every client might be different. The backtracking degree is represented by two hyperparameters, *backtracking region* and *backtracking time*. Backtracking region is an integer $r$, which means that the last $r$ tasks will appear in this backtracking process. Backtracking time is an integer $t$, which means there are $t$ backtracking rounds after an ordinary round. We apply the settings to all three datasets as the repetitive task simulation.

## A.2  BASELINES AND HYPERPARAMETER SETTINGS

TagFed is compared to PackNet+FedAvg, GLFC, FedWeIT and FedKNOW, which are briefly summarized as follows:

- *PackNet+FedAvg (PF) (Mallya & Lazebnik, 2018; McMahan et al., 2017):* This baseline is a combination of CL methods and FL methods. Concretely, we first pack multiple tasks into a model by iterative pruning and then average the parameters of the client models as federation.

- *GLFC (Dong et al., 2022):* GLFC is a state-of-the-art approach focusing on addressing the catastrophic forgetting problem in FCL. The key idea is to implement a class-aware gradient compensation loss and a class-semantic relation distillation loss to balance the forgetting of old classes.

- *FedWeIT (Yoon et al., 2021):* This method employs weighted inter-client transfer to address the issue of interference from irrelevant clients.

- *FedKNOW (Luopan et al., 2023):* This method ensures the prevention of catastrophic forgetting and mitigation of negative knowledge transfer by effectively combining signature tasks identified from the past local tasks and other clients' current tasks.

We now describe the standard implementation of TagFed, which is used throughout our experiments unless otherwise specified. The concrete parameter settings are as follows:

We employ ResNet-18 (He et al., 2016) as the backbone and extract feature maps after the second BasicBlock. The hyper-parameter $\alpha_s$ and $\alpha_c$ are set to 0.9 in Eq. 1 Eq. 2. $\beta_s$ and $\beta_c$ are set to 0.1. The distillation temperature is set to 5. We use SGD as the optimizer for training model parameters, and the learning rate is set to 0.2 with a momentum of 0.9. We use Adam as the optimizer for training masks, and the learning rate is set to 1e-4. The sparsity of each layer is set to 0.1 to hold all tasks into a model, with the purpose of making a fair comparison. All of the experiments are conducted for roughly 80 federating rounds to guarantee convergence. Finally, we run each experiment 3 times and average them as the reported results.

**Implementation details** All our experiments are simulated and conducted in a server that has 4 GeForce GTX 3090 GPUs, 48 Intel Xeon CPUs, and 128GB memory. We implement TagFed in Python with PyTorch, and all the experiments are conducted on a ResNet-18 architecture (He et al., 2016), which is pre-trained with the ImageNet dataset (Deng et al., 2009).

# B  THEORETICAL ANALYSIS

In this section, we provide theoretical analyses to support our framework. Specifically, we respectively analyze the effectiveness of the key modules in TagFed.

**Privacy analysis of information transfer.** In TagFed, the information transfer is a key step to enable local task learning and central knowledge aggregation. Although TagFed does not upload the raw data directly, it exchanges the feature maps obfuscated with designed noise as the message. Therefore, we attempt to analyze the privacy property of our framework. Concretely, we utilize the following Theorem to measure privacy.

**Theorem B.1.** *When an unbiased attacker tries to reconstruct the training data from the trained model $\hat{\mathbf{x}} = \mathrm{Att}\,(\mathbf{z}')$, the average mean-square error (MSE) of $\hat{x}$ is bounded by:*

$$\mathbb{E}\left[\|\hat{\mathbf{x}} - \mathbf{x}\|_2^2\right] \geq d/\operatorname{Tr}\left(\mathcal{I}_{z'}(\mathbf{x})\right) \tag{3}$$

*where $d$ is the dimension of $\mathbf{x}$ and $\mathrm{Tr}$ is the trace of a matrix.*

Here, $\operatorname{Tr}\left(\mathcal{I}_{z'}(\mathbf{x})\right)/d$ is called *diagonal Fisher information leakage*, or dFIL. dFIL is a strong indicator of privacy when it comes to input reconstruction attacks for split inference. For an unbiased attacker, dFIL indicates the reconstruction feasibility. Details can be found in (Guo et al., 2022).

In our work, TagFed allows clients to choose different variances of noise according to the client requirement. Specifically, each client can provide a privacy and utility requirement, which will be used to calculate the concrete noise to ensure the trade-off. Here we calculated the noise based on an empirical value (dFIL = 1) and introduced it into our training process to obtain values in the main text.

**Effectiveness analysis of group-wise knowledge aggregation.** In this part, we mainly explore whether the group-wise scheme contributes to knowledge aggregation. To begin with, we make the following commonly used assumptions.

**Assumption B.2.** The objective functions $F_1, F_2, ..., F_N$ in each device are all L-smooth: for all $\mathbf{v}$ and $\mathbf{w}$, $F_k(\mathbf{v}) \leq F_k(\mathbf{w}) + (\mathbf{v} - \mathbf{w})^T \nabla F_k(\mathbf{w}) + \frac{L}{2}\|\mathbf{v} - \mathbf{w}\|_2^2$.

**Assumption B.3.** The objective functions $F_1, F_2, ..., F_N$ in each device are all μ-strongly convex: for all $\mathbf{v}$ and $\mathbf{w}$, $F_k(\mathbf{v}) \geq F_k(\mathbf{w}) + (\mathbf{v} - \mathbf{w})^T \nabla F_k(\mathbf{w}) + \frac{\mu}{2}\|\mathbf{v} - \mathbf{w}\|_2^2$.

According to a related work (Zhao et al., 2018), the *weight divergence* among the uploaded information can affect the aggregation performance, which can be formally defined as the following Theorem.

---

**Algorithm 1** Pipeline of TagFed

---

**Operation**:

1: Distribute a shared model $M_{ini}$ to each client
2: **for** *i=1 to* $R$ **do**
3:     Conduct these steps in **Client side** and **Server side** sequentially
4: **end for**
5: Obtain the final improved model $M_{final}$

**Client side**:

1: For each client, train $M_{ini}$ with its own task sequence $S_t = \{s_1, s_2, ..., s_n\}$ based on our *traceable task learning*
2: Add noise to the feature maps of the hidden layers, upload them as well as the logits to the server
3: Wait for the predicted values sent back from the server
4: Conduct training based on Eq. 2

**Server side**:

1: Collect the uploaded feature maps $\{FM_1^i, FM_2^i, ..., FM_N^i\}(i = 1, 2, , , n)$ from $n$ clients
2: Feed them into different server models based on the uploaded client logits
3: Conduct training based on Eq. 1

---

**Theorem B.4.** *Suppose Assumption 1 and 2 hold the federation synchronization is conducted every H steps. The weight divergence after the m-th synchronization can be bounded by*

$$\|\boldsymbol{\Delta w_m}\| \leq \tag{4}$$

$$\sum_{k=1}^{N} \frac{sample^{(k)}}{\sum_{k=1}^{N} sample^{(k)}} (z^k)^T \|\boldsymbol{\Delta w_{m-1}}\|$$

$$+ \eta \sum_{k=1}^{N} \frac{sample^{(k)}}{\sum_{k=1}^{N} sample^{(k)}} \|p^k - p^{global}\| \sum_{j=1}^{H-1}$$

$$\left(z^k\right)^j \max_{i=1} \left\|\nabla_{\boldsymbol{w}} \mathbb{E}_{\boldsymbol{x}|y=i} \log f_i(\boldsymbol{x}, \boldsymbol{w})\right\|$$

*where $z^k = 1 + \eta \sum_{i=1}^{C} p^{(k)}(y = i)\lambda_{\boldsymbol{x}|y=i}$ and $C$ is the number of category. $p$ represents the data distribution.*

Based on the inequality, we can observe that the distribution heterogeneity among uploaded information has a large impact on the final divergence degree. Our group-wise scheme enables aggregation on the features of the same task, which significantly mitigates the distribution heterogeneity and benefits the final performance.

## C    ADDITIONAL EMPIRICAL RESULTS

Due to the page limitation of the main text, here we show our additional empirical results to further demonstrate the superiority of TagFed.

### C.1    PERFORMANCE ON THE DIFFERENT TASK SCALES

In the main text, we only test the performance on a fixed task scale. In this part, we attempt to explore the performance on different task scales. Figure 7 - Figure 9 summarize the results. From these tables, we can draw the following conclusions: (1) *PackNet+FedAvg* cannot achieve good performance no matter how many tasks are in the sequence. This demonstrates that traditional federated aggregation methods that modify the weights are not suitable to our scenario; (2) The performance of typical FCL baselines, *FedWeIT*, *FedKNOW*, and *GLFC* will degrade significantly as we increase the number of tasks. This suggests that it cannot cope with massive repetitive tasks; (3) Our TagFed maintains a stable and high accuracy for all task scales. In addition, we find that usually the more tasks we have, the superior performance we can achieve.

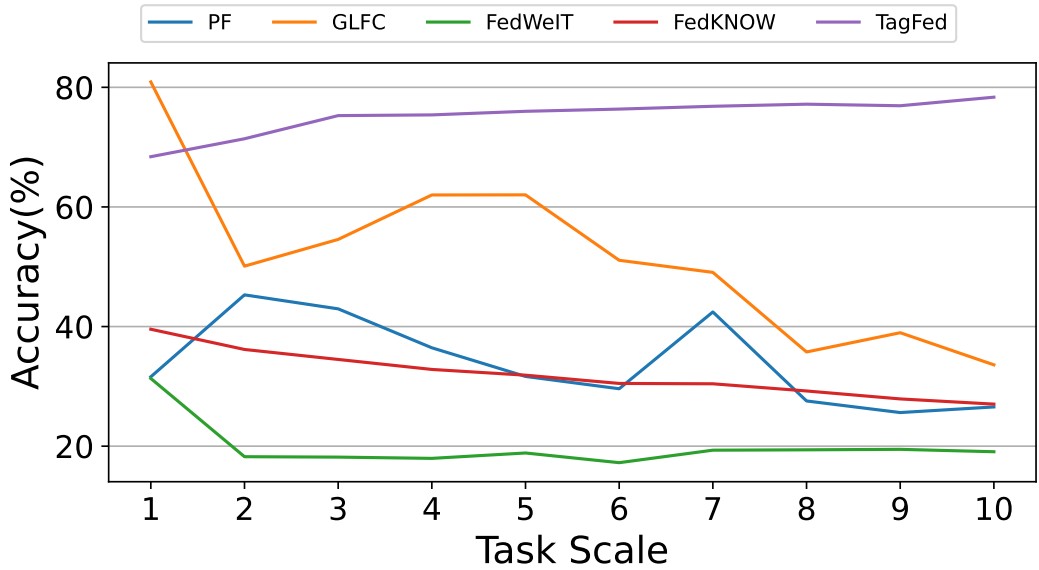

Figure 7: Performance of different task scales on CIFAR-100.

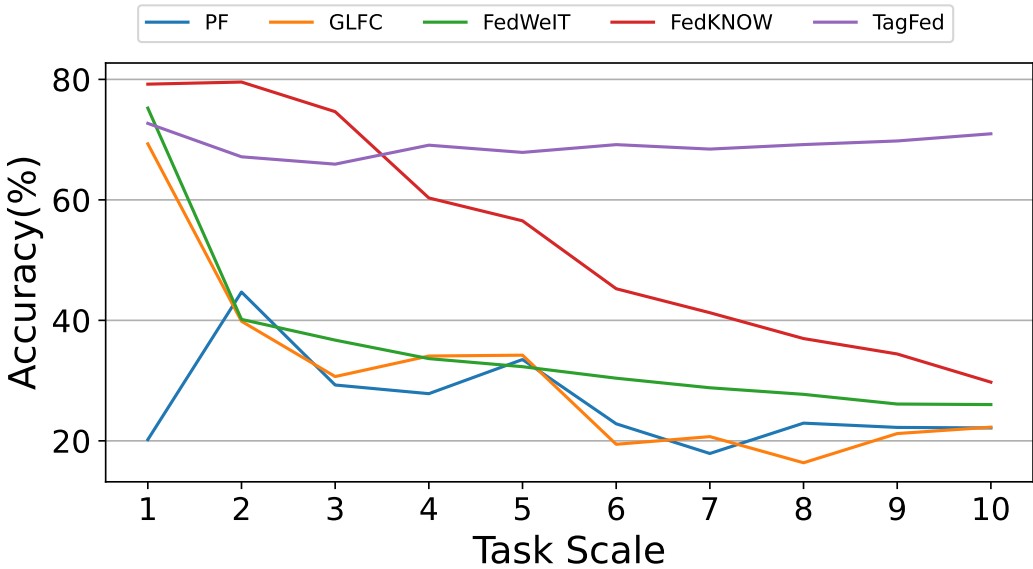

Figure 8: Performance of different task scales on ImageNetSubset.

## C.2 Task-specific performance on other datasets

Besides the performance of TinyImageNet in the main text, here we record the task-specific performance on ImageNetSubset and CIFAR-10, in order to test the generalization of our framework. Figure 10 and Figure 11 show the detailed accuracy performance. We can clearly see that TagFed can achieve consistent improvement on both datasets. Notably, The performance gap is higher in the middle tasks than others. We believe this is because the repeatability interference in these locations may have a higher impact on the baselines, leading to poor performance.

## C.3 Ablation study on other datasets

In the main text, we conduct the ablation study on CIFAR-100. Here we record the performance of other datasets, in order to see whether each module in TagFed is still effective. As shown in Table 5 and Table 6, both TTL and GKA play an important role in the final performance.

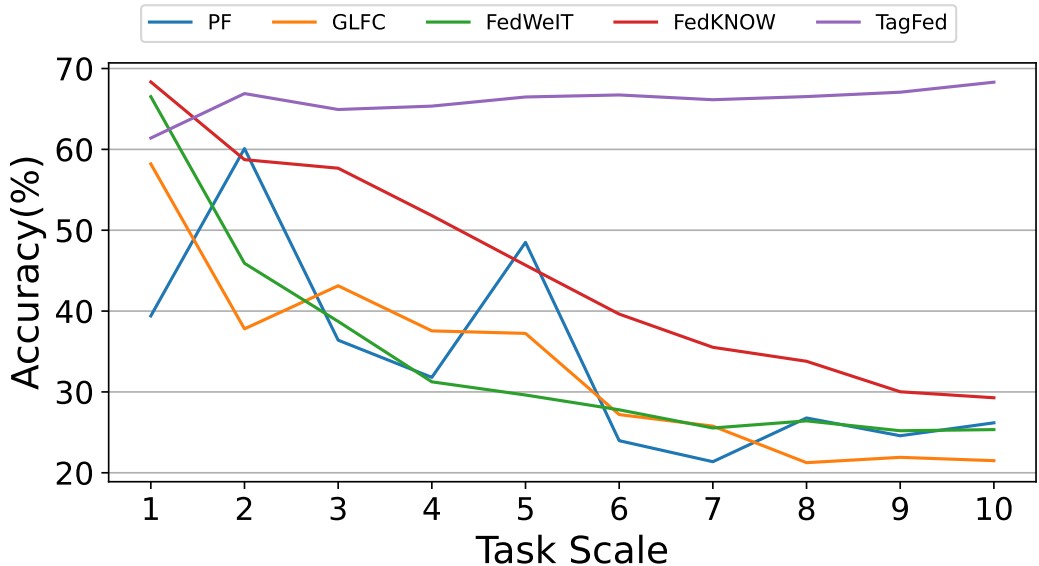

Figure 9: Performance of different task scales on TinyImageNet.

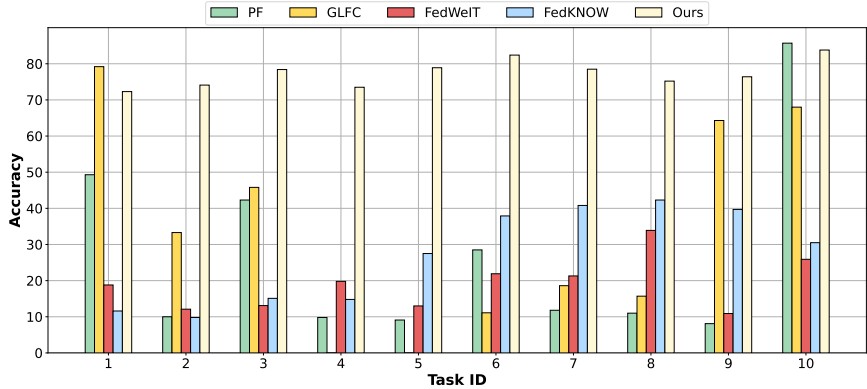

Figure 10: The task-specific performance on the CIFAR-100 dataset.

### C.4 NON-IID CONDITION AND REPEATABILITY DEGREE ON OTHER DATASETS

Figure 12 and Figure 13 show the results on ImageNetSubset and TinyImageNet, focusing on the non-iid condition and the repeatability degree. From these figures, we can find a similar conclusion to the results in the main text: TagFed can handle the non-iid condition well and will not be affected by the task repeatability degree.

Table 5: Effect of the proposed modules of different task numbers on ImageNetSubset. *Ours-w/oModule* denotes the performance of our framework without using the Module.

| Task Scale | 2 | 4 | 6 | 8 | 10 | Avg |
|---|---|---|---|---|---|---|
| *Ours-w/oTTL* | 52.15% | 40.00% | 34.48% | 30.60% | 30.72% | 37.59% |
| *Ours-w/oGKA* | 62.85% | 64.10% | 62.42% | 62.83% | 64.12% | 63.26% |
| *Ours* | **67.15%** | **69.08%** | **69.17%** | **69.18%** | **70.97%** | **69.11%** |

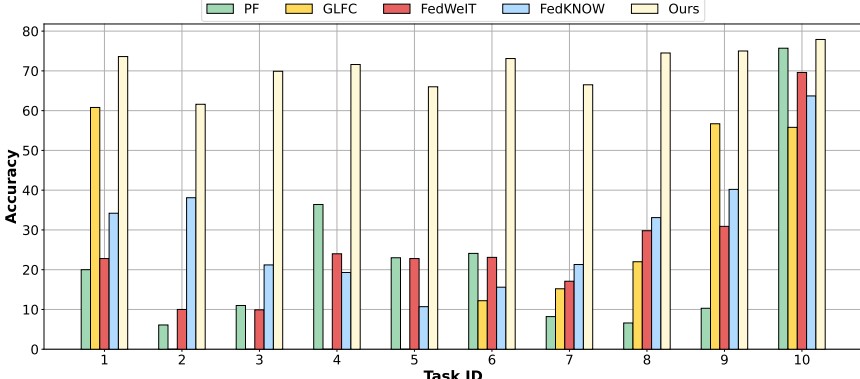

Figure 11: The task-specific performance on the ImageNetSubset dataset.

Table 6: Effect of the proposed modules of different task numbers on TinyImageNet. *Ours-w/oModule* denotes the performance of our framework without using the Module.

| Task Scale | 2 | 4 | 6 | 8 | 10 | Avg |
|---|---|---|---|---|---|---|
| *Ours-w/oTTL* | 57.10% | 42.40% | 36.30% | 35.60% | 36.54% | 41.59% |
| *Ours-w/oGKA* | 59.50% | 58.90% | 61.07% | 60.48% | 61.06% | 60.20% |
| *Ours* | **66.90%** | **65.35%** | **66.73%** | **66.53%** | **68.31%** | **66.76%** |

### C.5  NOISE-INTRODUCED PERFORMANCE ON OTHER DATASETS

In this subsection, we evaluate the noise-introduced performance on CIFAR-100 and ImageNetSubset and demonstrate the results in Table 7 and Table 8. From these tables, we can clearly observe that for any dataset, TagFed can exceed other baselines under the privacy-preserving situation.

### C.6  PERFORMANCE ON LARGE-SCALE CLIENTS

In above mentioned experiments, we only operate the FL system with 10 clients. To explore whether the proposed approach can be extended to a large-scale scenario, we add the number of clients to 30 and implement each method respectively. The experiments are conducted on CIFAR-100. As illustrated in Figure 14, TagFed surpasses other baselines by a large margin as we increase the number of tasks, which is consistent with our previous analysis.

## D  ALGORITHM AND REPRODUCTION

Algorithm 1 shows the whole pipeline of our proposed TagFed. For the client side, we mainly conduct our traceable task learning. For the server side, we focus on the group-wise knowledge aggregation. Note that the operations will be implemented sequentially until training convergence.

To ensure reproducibility, we have provided the overview of datasets and baselines in Appendix A.1 and Appendix A.2. Our experimental environment is presented in Section 5.1. We will make our code and other artefacts available to the community after the notification.

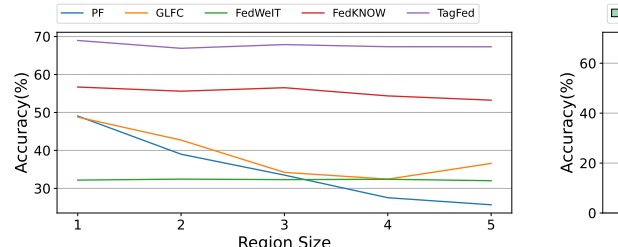
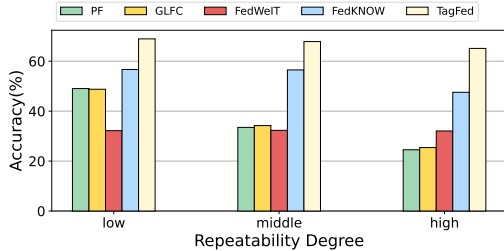

Figure 12: Results on the ImageNetSubset. Left: Performance on the non-iid condition. Right: Effect of the Repeatability Degree.

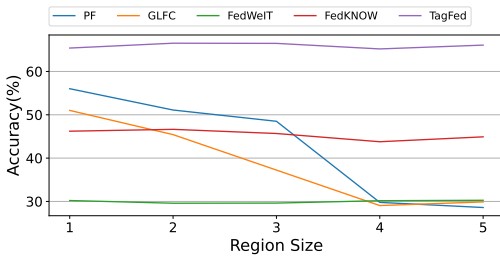
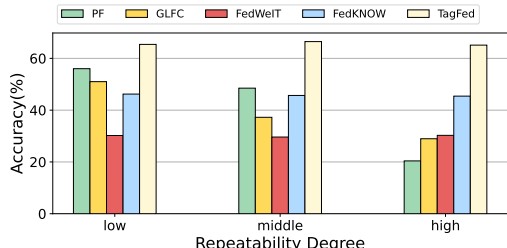

Figure 13: Results on the TinyImageNet. Left: Performance on the non-iid condition. Right: Effect of the Repeatability Degree.

Table 7: Results of the noise-introduced situation on CIFAR-100. Here *PF* denotes *PackNet+FedAvg*.

| Task scale | 2 | 3 | 5 |
|---|---|---|---|
| *PF* | 45.30% | 42.96% | 31.66% |
| *GLFC* | 50.10% | 54.57% | 62.02% |
| *FedWeIT* | 18.24% | 18.17% | 18.86% |
| *FedKNOW* | 36.17% | 34.50% | 31.87% |
| *Ours(1/dFIL = 1)* | 70.50% | 73.86% | 73.60% |
| *Ours(w/o noise)* | 71.40% | 75.27% | 75.99% |

Table 8: Results of the noise-introduced situation on ImageNetSubset. Here *PF* denotes *PackNet+FedAvg*.

| Task scale | 2 | 3 | 5 |
|---|---|---|---|
| *PF* | 44.70% | 29.27% | 33.50% |
| *GLFC* | 39.85% | 30.67% | 34.22% |
| *FedWeIT* | 40.16% | 36.72% | 32.31% |
| *FedKNOW* | 79.56% | 74.64% | 56.52% |
| *Ours(1/dFIL = 1)* | 66.70% | 63.90% | 64.10% |
| *Ours(w/o noise)* | 67.15% | 65.93% | 67.88% |

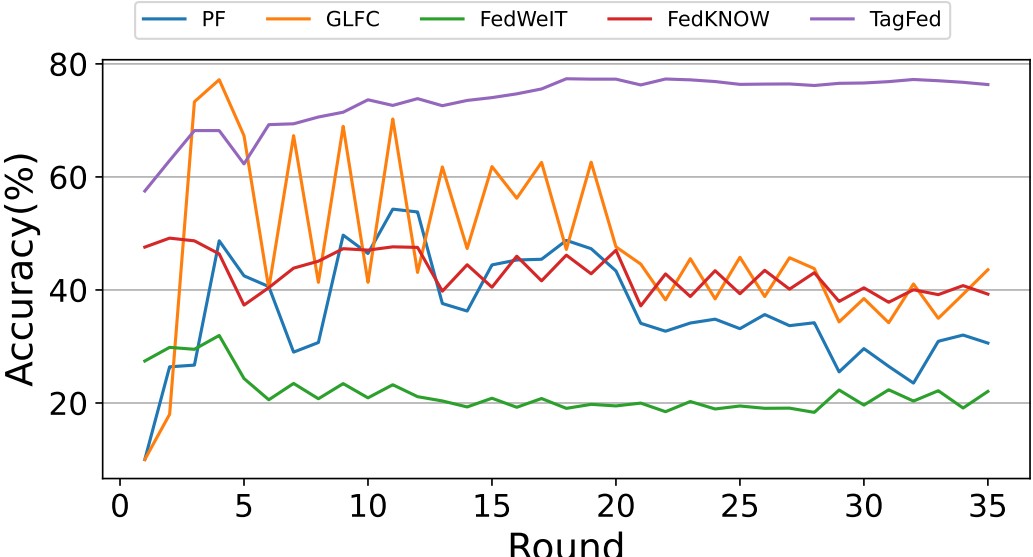

Figure 14: Performance on large-scale clients.

