# OpenReview forum: "Traceable Federated Continual Learning"
_ICLR.cc/2024/Conference — ICLR 2024 Conference Withdrawn Submission_

### Official Review · Reviewer_drQ8 · 2023-10-30

**Soundness:** 2 fair
**Presentation:** 3 good
**Contribution:** 3 good
**Rating:** 5
**Confidence:** 3

**Summary:**

The manuscript introduces a specific sub-category of federated continual learning where, tasks are repeatable and evoke the necessity to handle such conditions in practical scenarios.
With this setup in mind, the paper introduces TagFed, a framework that actively traces tasks and ensures the usage and optimization of dedicated sub-models for each task.
The authors use a novel benchmark to evaluate the new paradigm in federated continual learning and TagFed.

**Strengths:**

- The authors contextualize their work extremely well, highlighting important issues and assumption made in federated continual learning.
- Given the contributions, the experimental validation is very thorough.
- Detailed descriptions of the setup and highlights of important takeaways from the experiments offer a good reading experience to the user.

**Weaknesses:**

- TagFed uses weight-based pruning to obtain task-specific sub-models and then uses distillation to ensure that appropriate server and client models can distill and learn from one another. However, the specific type of clustering/aggregation performed and at which point in the server and client they are performed remain slightly obscure.
- Since TagFed maintains a copy of weights for each task, with an increase in the number of tasks there would be a large memory overhead within each client, especially if we consider the worst case scenario of every client observing all the available set of tasks.
- The notion of cross-entropy loss at the server level is confusing since there was no clear definition of a public dataset at the server level.
- Since traceability is argued to be critical to the functioning of TagFed under the proposed FCL setting, explicit evaluation of the task detection mechanism against GLFC's "Task Transition Detection" measure would add a more straightforward comparison and deepen the discussion on the importance of this measure.

**Questions:**

- Could the authors detail the exact (a) nature of clustering/identification of sub-tasks, and (b) whether are they performed at both the server and client side?
- Given the repetitive pruning mechanism used to obtain task-specific sub-models, could the authors discuss whether they observe a drop in effective capacity with an increase in the number of classes/tasks?
- Could the authors confirm whether the intuition of memory overhead, as stated in the weaknesses, matches the expectation from TagFed with an increase in the number of tasks? If so, do you have a solution to avoid storing multiple such copies?
- Does TagFed use a public dataset at the server level? If not, how is ground-truth obtained for the cross-entropy loss? If there is a public dataset, could you describe the specifics of which dataset was used and how?
- Could the author's compare their approach against GLFC's "Task Transition Detection" measure, purely from a task detection capability standpoint?
- From an ablation standpoint, could the authors provide more in-depth descriptions of how w/o TTL and w/o GKA experiments are set up?

---

### Official Review · Reviewer_oWSe · 2023-10-31

**Soundness:** 2 fair
**Presentation:** 2 fair
**Contribution:** 2 fair
**Rating:** 3
**Confidence:** 3

**Summary:**

The authors developed an approach called traceable federated continual learning (TFCL) for federated learning settings where the model sees a few tasks sequentially with repetition. The TFCL approach is composed of pruning-based sub-model generation and server-client knowledge distillation.

**Strengths:**

This paper studies an interesting setting: continual learning with repetitive tasks.

**Weaknesses:**

1. If I understand Section 4.2 correctly, the traceable augmentation simply duplicates the model a few times and lets each copy deal with a task. This seems to be a baseline implementation.
2. If we decide to use separate models for each task, the pruning-based sub-model generation does not seem necessary. Although pruning may reduce computation and storage costs, that is not closely related to the problem in this paper.
3. If the pruning-based sub-model generation is not necessary, there is no need to use a knowledge distillation framework to aggregate models. The knowledge distillation framework enables the aggregation of heterogeneous models. However, the model heterogeneity is introduced by an unnecessary sub-model generation operation.

**Questions:**

What does "$p_q = p_i = ... = p_j$" in Section 4.2 imply?

---

### Official Review · Reviewer_1NM3 · 2023-11-01

**Soundness:** 2 fair
**Presentation:** 2 fair
**Contribution:** 2 fair
**Rating:** 5
**Confidence:** 3

**Summary:**

This paper focused on federated continual learning, and proposed Tractable Federated Continual Learning (TFCL), where repetitive tasks can be accurately and effectively traced and processed to boost the performance. In particular, the proposed TagFed framework has been implemented to achieve TFCL through feature tracing, submodel augmentation and group-level knowledge federation. Experimental results were provided to validate the performance of TagFed.

**Strengths:**

This paper proposed TagFed for Tractable Federated Continual Learning (TFCL). Specifically, TagFed includes feature tracing, submodel augmentation and group-level knowledge federation. Experimental results were provided to validate the performance of TagFed.

**Weaknesses:**

1. In order to implement TagFed, it needs to check whether or not the current task is a repetitive task. This is done through tracing and connecting to previous tasks. As a result, there is a natural tradeoff between performance and costs in terms of memory. If TagFed stores the previous tasks over a very long period, it will benefit TagFed which tasks advantage of repetitive tasks. However, this will need a lot of memory as well. Though there are some discussions in this paper. It is not clear to the reviewer how TagFed fully addresses this issue. Do you leverage a “time-window” to determine “the length of the time periods” for the previous tasks to be stored? Is it a sliding window? How to determine the length of such a window? A fixed or adaptive value of the window size?
2. Following the above question, in Table 3, the authors present some numerical results on the complexity of TagFed and baselines. There is no clear statement about the experimental settings (neither in the appendix), and hence it is not easy to justify how “good” or “bad” these values. For instance, 11.7MB can be a very large or small cost depending on the system configuration. No memory cost is provided.
3. In Section 3.1, the authors claimed that recent works did not consider the data repeatability in a task sequence. However, cannot they still be applied in the presence of data repeatability? These recent works are designed for general task sequences which include the repeatable ones.
4. This paper leverages several state-of-the-art datasets for image classification tasks, which are widely used in the FL setting. However, in general, these datasets do not this data repeatability property. How do you generate the data repeatability cases from these datasets?
5. In Section 3.2, “For a certain time”, this is a quite myopic description. How frequent is the repeatability? what’s its impact on the performance?
6. To achieve the incremental training for non-repetitive new tasks, a threshold is needed. However, the reviewer did not find information on how to determine this threshold, what’s impact on the performance, do you need to tune this value? No ablation study was provided.
7. A minor question: for the traceable augmentation for repetitive new tasks, it seems that it requires the tasks to repeat at least twice (therefore the value of j can be meaningful). The implementation indeed requires the information of task p_j. However, what will happen if there is no such j? i.e., only repeat once?
8. The proposed TagFed is most heuristic based method without strong theoretical performance guarantee (E.g., convergence analysis).

**Questions:**

See the comments above in weakness.

---

### Official Review · Reviewer_MVcU · 2023-11-01

**Soundness:** 2 fair
**Presentation:** 3 good
**Contribution:** 2 fair
**Rating:** 5
**Confidence:** 3

**Summary:**

Authors propose a new paradigm, namely Traceable Federated Continual Learning (TFCL), aiming to cope with repetitive tasks by tracing and augmenting them. Following the new paradigm, they further develop TagFed, a framework that enables accurate and effective Tracing, augmentation, and Federation for TFCL.

**Strengths:**

1. This paper proposes a new paradigm: Traceable Federated Continual Learning (TFCL), which copes with repetitive tasks by tracing and augmenting them.

2.  Following the new paradigm, this paper develops a framework that enables accurate and effective tracing, tracing, augmentation, and federation for TFCL.

3. Results are promising

**Weaknesses:**

I wonder the motivation of this setting. Leave federated learning alone, let us focus on the continual learning part, I wonder if there are any previous works in the continual learning research community that study the case proposed in this paper? (i.e., the upcoming task data are NOT completely different from previous tasks). From my understanding, this setting makes the modification from the perspective of continual learning.

If there are, have you combined them into your setting to setup a series of simple baselines? For instance, combining federated algorithms like FedAVG with the continual learning algorithms.

Also, have you tried to compare with recent related works on FCL? For insrance [1,2], which do not required task identifiers. Can you clarify and discuss about them?

Missing related works:
[1] S Babakniya et al., Don't Memorize; Mimic The Past: Federated Class Incremental Learning Without Episodic Memory
[2] D Qi et al., Better generative replay for continual federated learning

**Questions:**

Please refer to Weaknesses.